# A novel approach for longitudinal analysis of serum biomarkers of joint metabolism and knee injury in military officers

Liubov Arbeeva[1]*, Virginia B. Kraus[2], Amanda E. Nelson[1,3], Maryalice Nocera[4], Leigh F. Callahan[1], Richard F. Loeser[1,3], Kenneth L. Cameron[5], Jesse R. Trump[5], Stephen W. Marshall[4,6], Yvonne M. Golightly[1,6,7]

1 Thurston Arthritis Research Center, University of North Carolina at Chapel Hill, Chapel Hill, North Carolina United States of America, 2 Duke Molecular Physiology Institute and Division of Rheumatology, Department of Medicine, Duke University School of Medicine, Durham, North Carolina United States of America, 3 Division of Rheumatology, Allergy and Immunology, University of North Carolina at Chapel Hill, Chapel Hill, North Carolina, United States of America, 4 Injury Prevention Research Center, University of North Carolina at Chapel Hill, North Carolina United States of America, 5 John A. Feagin Jr. Sports Medicine Fellowship, Keller Army Hospital, United States of America Military Academy, West Point, New York, United States of America, 6 Department of Epidemiology, University of North Carolina at Chapel Hill, North Carolina, United States of America, 7 College of Allied Health Professions, University of Nebraska Medical Center, Omaha, Nebraska, United States of America

* liubov_arbeeva@med.unc.edu

## Abstract

### Purpose

To investigate the longitudinal relationships between serum biomarkers of joint metabolism, knee injury, and Knee Injury and Osteoarthritis Outcome Score (KOOS) using novel methodologies.

### Methods

Data were collected from military officers who enrolled as cadets between 2004–2009, with follow-up conducted between 2015–2017. Analyses included 234 officers who had no history of knee ligament/meniscal injury at the time of military academy matriculation, had serum biomarker measurements at matriculation and graduation, demographic data, and KOOS assessment at follow-up. Biomarkers included Collagen Type II (C2C) and Type I and II (C1,2C) collagenase-generated cleavage epitopes, C-terminal propeptide of Type II collagen (CPII), and C- and N-terminal telopeptides of type I collagen (CTX and NTX). Angle-based Joint and Individual Variation Explained (AJIVE) was used to determine demographic determinants of biomarker levels and individual modes of variation specific to biomarker levels at matriculation and graduation, stratified by sex.

or otherwise used by anyone for any lawful purpose. The work is made available under the Creative Commons CC0 public domain dedication.

**Data availability statement:** Minimal data for this study cannot be shared publicly because there are legal restrictions given that this research was conducted in U.S. Armed Forces. Data are available upon request from the Institutional Review Boards of the University of North Carolina at Chapel Hill (UNC-CH 13-0059) and the Keller Army Community Hospital (KACH 15-007) via email (irb_questions@unc.edu), postal mail (104 Airport Drive, Suite 2100, CB#7097, Chapel Hill, NC 27599), or telephone ((919) 966 3113), for researchers who meet the criteria for access to confidential data. All study data are securely stored on UNC-protected servers, and in the event that all authors become unavailable, UNC IT personnel will maintain access and provide data upon request.

**Funding:** Financial support was provided by the National Institute of Arthritis and Musculoskeletal and Skin Diseases (P60AR064166, P30AR072580). The funder provided support in the form of salaries for authors but did not have any additional role in the study design, data collection and analysis, decision to publish, or preparation of the manuscript. The specific roles of these authors are articulated in the 'Author Contributions' section.

**Competing interests:** AEN reports honoraria from Novo Nordisk, MedScape Education, and CCR-West 2024, support for travel from NYU Langone, Hospital for Special Surgery, Osteoarthritis Research Society International, and American College of Rheumatology as well as grant funding from NIH/NIAMS not related to this work (R01AR078187; R01AR080742; K24AR081368; R01AR077060; R01AR080733). YMG reports grant funding from NIH/NIAMS not related to this work (R34AR083077; R01AR078187; R01AR080742; R01AR080733; R56AR080060; R21AR080309). SWM reports grant funding from NIH/NIAMS not related to this work (R01 AR050461). This does not alter our adherence to PLOS ONE policies on sharing data and materials.

## Results

We confirmed known associations of joint metabolism biomarkers with age in both sexes and with smoking in males. Matriculation biomarker data in males suggested a protective biomarker profile characterized by high cartilage synthesis and low cleavage of type I and II collagen in association with healthy KOOS scores at follow-up. CPII measured at matriculation was negatively associated with incident injuries after adjustment for smoking status (p = 0.03, logistic regression), confirming results from AJIVE.

## Conclusion

These exploratory analyses suggest that CPII alone, or in combination with other joint metabolism biomarkers, may help identify individual risk of knee injury.

## Introduction

Osteoarthritis (OA) is the most common joint disease, has no cure, and worsens over time, leading to chronic pain and disability [1]. Globally, OA prevalence has increased by more than 130% over the past 30 years and is projected to rise by at least another 60% by 2050 [2]. Previously considered a simple "wear and tear" condition of older adults, OA is now recognized as a multifactorial, complex disease that can include structural changes, impaired physical function, and symptoms (e.g., pain) with a pronounced inflammatory component, often linked to metabolic dysfunction and increased biomechanical loading [3,4]. Notably, approximately 12% of all symptomatic OA cases are related to prior injury, with the knee being the most commonly affected joint [5,6]. After major injury, knee OA can develop relatively quickly (i.e., within the subsequent 10 years [7,8]) and can contribute to the occurrence of obesity and other comorbid conditions. The risk of symptoms, total knee replacement at a younger age, and subsequent need for revision surgery are higher in active individuals such as military officers compared to the general population [9–15]. All these factors have a dramatic impact on quality of life, physical performance, and the ability to perform work and military duties in affected individuals [16].

In contrast to knee OA occurring without a known joint injury, the initiation of injury-related knee OA is more precisely identifiable, providing a unique window during which to observe the earliest changes in joint metabolism before the onset of symptoms and irreversible pathological changes. Soluble biomarkers may provide a valuable tool for detecting these early molecular changes, supporting earlier diagnosis and informing preventive strategies [17].

Numerous biomarkers have been investigated in human and animal studies of injury-related knee OA, yet none are currently recommended for monitoring knee OA development in humans [17]. Key gaps in biomarker research for primary and injury-related OA include which biomarkers, or their combination, best predict disease trajectory, the optimal timing for assessment, and how to distinguish beneficial/compensatory (i.e., those due to the early inflammatory response to

joint injury) from pathological changes. There is also limited understanding of irreversible damage onset, differences between traumatic and non-traumatic OA, and the role of systemic factors and comorbidities. Furthermore, biomarker levels can be affected by multiple factors, including age, physical maturity, gender, body mass index (BMI), and recent physical activity [18]. Many human biomarker studies lack healthy, age-matched controls, making it difficult to distinguish disease-specific changes. Including such controls, along with longitudinal biomarker assessments, is essential for improving the early detection and prevention of OA. Several military-specific factors may advance the field of biomarkers for injury-related knee OA. Military cohorts, particularly cadets with knee injuries in the early stages of OA development, offer a unique opportunity to discover blood-based biomarkers for early diagnosis in healthy, young, and active individuals to ensure that discoveries translate into meaningful benefits for populations at greater risk. Additionally, the rapid timeline from knee injury to radiographic knee OA (before the age of 30 years [19]) among military personnel provides the opportunity for longitudinal assessment of biomarker levels to detect meaningful changes in OA progression, particularly if the biomarkers are collected pre- and post-injury. Novel methodologies, such as machine learning techniques, can provide new insights into relationships of multiple biomarkers and various demographic and clinical data to study biological processes occurring from injury to knee OA, while taking into account other processes in the body that may not be directly related to OA development. Therefore, the purpose of this study was to utilize a machine learning approach to examine the longitudinal relationships between serum biomarkers of joint metabolism and participant characteristics in a cohort of physically active military officers, first among individuals without baseline injury to examine risk for injury, and second, among those with injury before, during, and after the academy to explore risk for knee problems consistent with early OA.

## Methods

### Study design

This study is a secondary analysis of data collected from military officers enrolled in the Joint Undertaking to Monitor and Prevent Anterior Cruciate Ligament Injury (JUMP-ACL), a prospective cohort study designed to examine risk factors for anterior cruciate ligament (ACL) injuries [20]. The overall study design and participants flow are shown in Fig 1.

### Participants (Study population)

Between 2004 and 2009, a total of 6,452 military cadets/midshipmen (39% female) were enrolled in JUMP-ACL at the time of their matriculation to the Air Force, Naval, or Military Academies. During 2015−2017, 4,643 JUMP-ACL participants were invited by email to participate in a follow-up study called JUMP-ACL OA to examine the course of knee OA after injury and to identify key biomarkers associated with OA progression. The goal was to re-consent and enroll two appropriately equally sized groups of participants for JUMP-ACL OA: a group with a history of knee ligament and/or meniscal injuries at the time of JUMP-ACL OA enrollment, and the non-injured group without history of traumatic knee injury. The eligibility criteria and other details have been described previously [19]. In total, 479 officers with and without a history of traumatic knee injuries (e.g., ACL injury, meniscus injury) were consented using a multi-part sequential online consenting form and enrolled in JUMP-ACL OA between 5th March 2015 and 7th November 2017. Among these 479 officers, 437 completed a questionnaire between 2015−2017 and provided information on demographic characteristics, injury occurrence during and after the time in the academy, and the Knee Injury and Osteoarthritis Outcome Score (KOOS), a self-report measure of knee problems (e.g., symptoms, disability, reduced quality of life). For the JUMP-ACL OA study, injured and non-injured participants were consented and enrolled. This analysis included a subset of participants from the JUMP-ACL OA study without knee ligament/meniscal injury at matriculation (N = 350) who had available biomarker measurements at both matriculation and graduation (N = 234). This study was approved by the Institutional Review Boards of the University of North Carolina at Chapel Hill (UNC-CH 13−0059) and by the Keller Army Community Hospital (KACH 15−007).

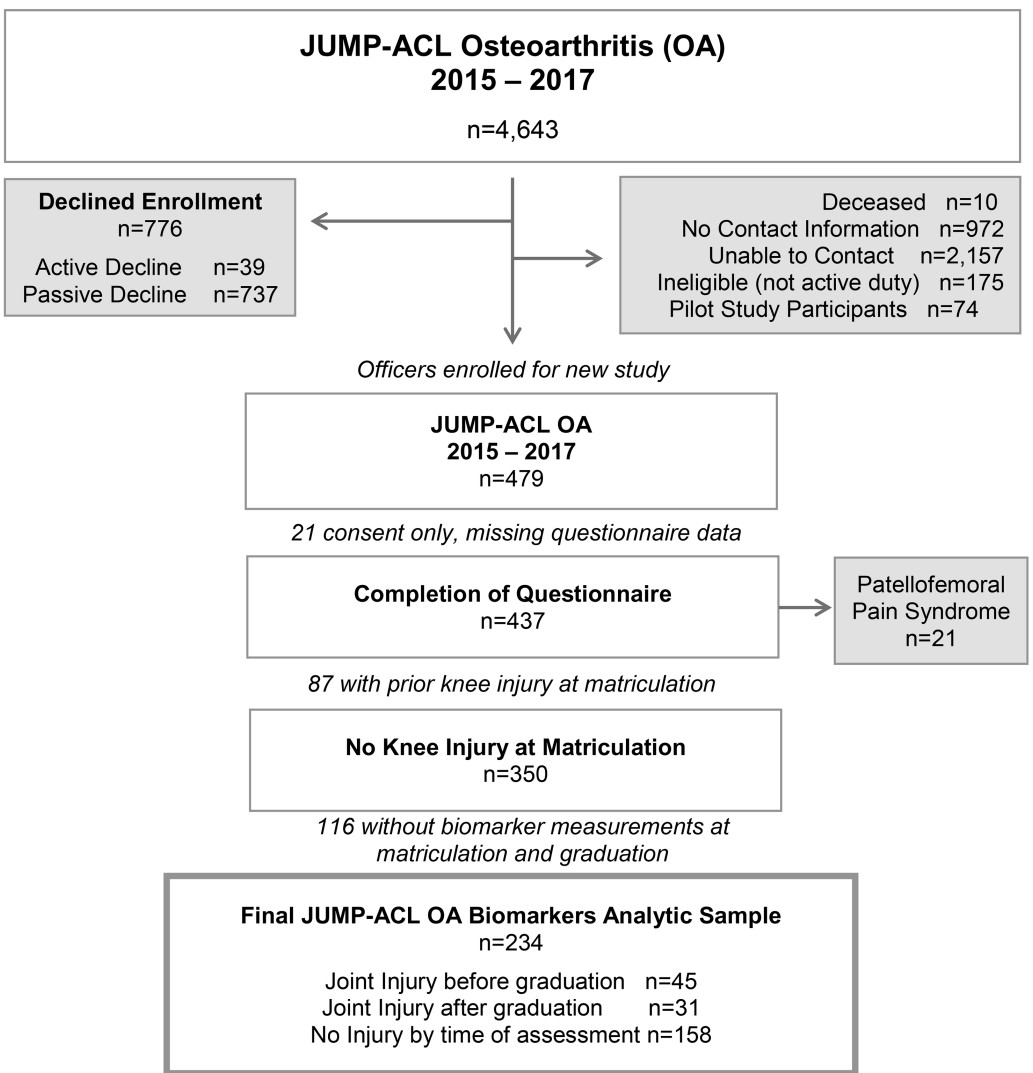

**Fig 1. Participants available for JUMP-ACL osteoarthritis biomarker analysis.**

## Measurements

**Demographics, Smoking Status, and BMI.** Gender, race, ethnicity, and smoking status were self-reported. Age and (BMI) at matriculation and graduation were available from the JUMP-ACL data set.

**Biomarkers.** Serum samples were obtained from the Department of Defense Serum Repository (DoDSR). For this study, samples were originally collected from cadets at two time points: upon matriculation to a military service academy

(in this study, between June 30 and July 7th, from 2004 to 2009) and again just prior to graduation and commissioning as a military officer (between May 1 and May 30, from 2008 to 2014) [21,22]. Samples were required to have at least 0.5 milliliter in volume to be eligible for research purposes. Samples were transferred from the DoDSR to the Biochemistry Lab in the Department of Chemistry and Life Science at West Point on 8th January 2019 and stored at −80 degree Celsius in an Ultra Low Temperature Freezer until analysis. For all samples from the DoDSR shipment, the manifest was used to create a masked layout of planned testing based on the selected assays, which included manufacture standards and the DoDSR samples. At the time this study was initiated, biomarkers were selected based on the availability of commercial assay kits and the presence of human data supporting their relevance to one or more categories within the BIPEDS classification system (Burden of disease, Investigational, Prognostic, Efficacy, Diagnostic, Safety) [23].The selected assays included Serum levels of Collagen Type II (C2C) and Type I and II (C1,2C) collagenase-generated cleavage epitopes, C-terminal propeptide of Type II collagen (CPII), and C- and N-terminal telopeptides of type I collagen (CTX and NTX). Based on BIPEDS (B=burden of disease, I=investigational, prognostic, E=efficacy of intervention, D=diagnostic, S=safety) osteoarthritis biomarker classification system and supporting literature, the selected biomarkers span multiple categories: **C2C** (B, D, E, S), **C1,2C** (B), **CPII** (B, P), **CTX** (P), and **NTX** (B, P, E, D) [23–28]. Commercially available kits (C2C; C1, 2C; CPII; CTX; NTX) were used for all assays, and analyses were conducted according to manufactures protocol. Assay kits were removed from cold storage and brought to room temperature on the bench top for 60 minutes. Samples were removed and brought to room temperature at the same time. Once room temperature was achieved, samples, standards and reagents were then placed into the EPmotion5075 (Eppendorf) and aliquoted to the target plate, with duplicate testing for each unknown sample, pending sample volume, using a custom robotic protocol for each assay to increase pipetting accuracy. The assays were measured using a BioTek Synergy microplate reader to determine optical density (OD) of the manufacture standards. The OD of the manufacture standards for each assay was then used to create a standard curve plot. The average OD of the duplicate samples was calculated and plotted along the standard curve to determine concentration. Samples with greater variability (2.5 standard deviation from the mean, or no concentration) were retested to ensure accuracy. During analysis, technicians were blinded to all participant information other than a unique specimen identifier number.

## Outcomes assessment

**Knee injury.** ACL and knee injuries of comparable severity that were experienced as a military cadet, or as an active duty officer, underwent surgical repair and post-surgical rehabilitation from experienced orthopedic and post-surgical team, with the goal of restoring full pre-injury function to the fullest extent possible. If surgery is not an option, then injuries were managed and rehabilitated using best possible surgical techniques.

**Knee problems.** The KOOS items were administered between 2015 and 2017, therefore they were obtained at a variable number of years after injury, with an average of 7 years post-injury. KOOS scores were computed for pain (9 items), function in daily living (short version, 7 items), symptoms (7 items), quality of life (4 items), and function in sport and recreation (5 items). Knee problems consistent with early stage OA was defined using Luyten criteria (e.g., scoring ≤85% in at least 2 out of 4 KOOS categories with function in sport and recreation excluded) [29]. At preinjury baseline, participants provided information regarding knee ligament injuries prior to matriculation. During their time at the academy, cadets were monitored for knee injuries via local injury surveillance systems, operating room records, and the Defense Medical Surveillance System (DMSS). Post-graduation injury information was self-reported via questionnaire at the time of follow-up.

## Statistical analysis

Descriptive statistics were calculated for participant demographics, including means and standard deviations for continuous variables, and frequencies and percentages for categorical variables. To account for association of biomarkers with age, we used both raw measurements of biomarkers and age-adjusted values computed as the residuals from linear multiple regression models of each biomarker on age, fitted separately in females and males. Standardization of continuous

variables (i.e., age, BMI, biomarker levels at matriculation and graduation, raw and age-adjusted) was performed with a shifted log transformation to ensure that distributions were close to normality and to bring variables to the same scale [30]. Biomarker and demographic data were divided into three sets of measurements, called blocks: 1) transformed and standardized biomarker levels at matriculation, 2) transformed and standardized biomarker levels at graduation, and 3) demographics (ethnicity, race, and age), smoking status, and BMI.

The analysis framework was based on the Angle-based Joint and Individual Variation Explained (AJIVE) algorithm aimed at finding shared (e.g., common to all data blocks) and individual (e.g., specific to each block) modes of variation to reveal a multi-dimensional data structure [31]. Similar to Principal Components Analysis (PCA), AJIVE is based on a decomposition of the data into modes of variations, using loadings and score matrices for interpretation. The loadings are the weights for each original variable when calculating the principal component (PC). The loading matrix provides variable level information and can be used to identify the importance of each variable to each of the PCs. The score matrix contains subject level information, which represents the original data (i.e., the original variables) with a reduced number of variables, or PC scores, while retaining most of the important information. Unlike PCA, AJIVE can work with multiple data blocks and provides two types of modes of variations (i.e., shared variation between blocks and individual variation for each block). Because biomarkers and demographic data, represented by three blocks, were measured on the same set of participants, it was possible to have common scores. However, common loadings were not possible. For example, a biomarker can be elevated at matriculation due to younger age, but not at graduation, which can be determined by different loadings for three variables from different blocks (i.e., biomarker at matriculation, graduation, and age). Therefore, the loadings indicate which variables in each block vary together or individually. AJIVE provides two sets of loadings (shared and individual) for each original variable. Similarly, two sets of principal component scores are computed for each individual.

Applying AJIVE to the data blocks of demographic and biomarker data collected over time allows us to answer several research questions. First, serum biomarker levels can differ by multiple factors, including gender, race, ethnicity, BMI, and age [32]. The component(s) of shared variation may reflect specific patterns in biomarker levels that are associated with age at matriculation and graduation, smoking status, and/or demographic characteristics. This will not only confirm previous knowledge of these relationships, but also remove the strong influence of demographic characteristics on biomarker levels and allow the discovery of other associations that may be due to injury rather than demographics. For example, individual variations of biomarkers measured at matriculation are not related to demographic data and biomarkers measured at graduation. Therefore, the individual structure of the matriculation block may suggest biomarkers that can be used singly or in combination, to identify individuals at high risk of injury among those who were injury-free at matriculation.

In the present study, by describing the components of shared variation, we determined whether serum biomarker levels varied by demographic characteristics and smoking status. In our primary analysis, among individuals without prevalent injuries at matriculation, we explored the association between individual variations of matriculation block and injury during and after the academy (e.g., after matriculation) to determine whether serum biomarker levels can identify individuals at high risk of injury. Additional exploratory analyses were performed on individuals excluded from the main analysis due to prevalent injuries at matriculation combined with those from the main analysis who sustained an injury during and after the academy (e.g., individuals injured before, during, and after the academy) to explore whether biomarker levels identify individuals at high risk of post-injury symptoms. Because there are known differences in serum biomarker levels by sex, analyses were performed separately by sex.

## Results

### Participant characteristics

Our sample comprised 234 participants without a prior knee meniscal or ligament injury reported at matriculation (149 males, 85 females; mean age at matriculation 18.7±0.9 years). A total of 43 participants (18%) had injuries between

matriculation and graduation and 18 (8%) after graduation. The descriptive characteristics of study participants are presented in Table 1.

## Distribution of biomarker levels

The distributions of raw biomarker values at matriculation and graduation are shown in Table 1 for the total sample and females and males separately. Visual inspection of the density plots reported in the Supplementary Materials (**S1 Fig**) confirmed that shifted-log transformations of the biomarker concentrations produced close-to-normal distributions.

## AJIVE analysis of data from participants without knee ligament/meniscal injury at matriculation

**Shared variation.** In the 149 males, we found one direction of shared variation between three data blocks (S2 Fig) characterized primarily by older age at both matriculation and graduation. To eliminate the strong effect of age and to focus instead on the effect of other demographic characteristics, we used age-adjusted values of biomarkers in the first two blocks and removed age from block 3, which enhanced the effect of other demographic characteristics (**Fig 2**). This single shared direction of variation – characterized by lower levels of cartilage synthesis and bone turnover, clinically by higher BMI, less smoking, and Hispanic ethnicity, was consistent with known associations of joint metabolism biomarkers with age and smoking.

Similarly, in females (N = 85), we found one direction of shared variation (**Fig 3**). However, there were no shared variations between three data blocks when biomarker levels were replaced with age-adjusted values, which indicates that all

**Table 1. Demographic and clinical characteristics of 234 participants without prevalent injuries at matriculation.**

| Participant characteristics | Total (N = 234) | Females (N = 85) | Males (N = 149) |
|---|---|---|---|
| *White, n (%)* | 199 (85%) | 70 (82%) | 129 (87%) |
| *Black or African American, n (%)* | 10/234 | 5 (6%) | 5 (6%) |
| *Hispanic Ethnicity, n (%)* | 15 (6%) | 7 (8%) | 8 (5%) |
| *Mean age (SD) at matriculation, in years* | 18.72 (0.88) | 18.63 (0.99) | 18.77 (0.83) |
| *Mean age (SD) at graduation, in years* | 22.65 (0.90) | 22.60 (0.99) | 22.68 (0.85) |
| *Mean BMI (SD) at matriculation, kg/m$^2$* | 25.23 (3.00) | 23.60 (2.63) | 26.15 (2.81) |
| *Knee injury during academy, n (%)* | 43 (18%) | 14 (16%) | 29 (19%) |
| *Knee injury after academy, n (%)* | 18 (8%) | 6 (7%) | 12 (8%) |
| **Biomarker levels at matriculation,** *mean (SD)* | | | |
| *CTX, ng/mL* | 0.20 (0.14) | 0.16 (0.11) | 0.22 (0.15) |
| *NTX, nM BCE/L* | 20.65 (9.70) | 15.90 (6.08) | 23.36 (10.34) |
| *CPII, ng/ml* | 240.60 (169.25) | 268.65 (228.29) | 224.60 (121.67) |
| *C1,2C, ng/mL* | 230 (650) | 300 (1070) | 190 (130) |
| *C2C, ng/ml* | 50.72 (13.85) | 48.91 (12.24) | 51.76 (14.63) |
| **Biomarker levels at graduation,** *mean (SD)* | | | |
| *CTX, ng/mL* | 0.24 (0.18) | 0.17 (0.12) | 0.29 (0.19) |
| *NTX, nM BCE/L* | 18.37 (7.73) | 15.14 (5.74) | 20.21 (8.12) |
| *CPII, ng/mL* | 317.45 (283.34) | 331.88 (417.64) | 309.22 (164.93) |
| *C1,2C, ng/mL* | 280 (640) | 340 (1060) | 250 (80) |
| *C2C, ng/mL* | 62.27 (31.35) | 61.92 (31.89) | 62.47 (31.14) |

*BMI:* Body Mass Index; *CTX:* C-terminal telopeptides of type I collagen; *NTX:* N-terminal telopeptides of type I collagen; *CPII:* C-terminal propeptide of Type II collagen; *C1,2C:* Collagen Type I and II collagenase-generated cleavage epitopes; *C2C:* Collagen Type II collagenase-generated cleavage epitopes.

Missing data: *Hispanic ethnicity* – 2.

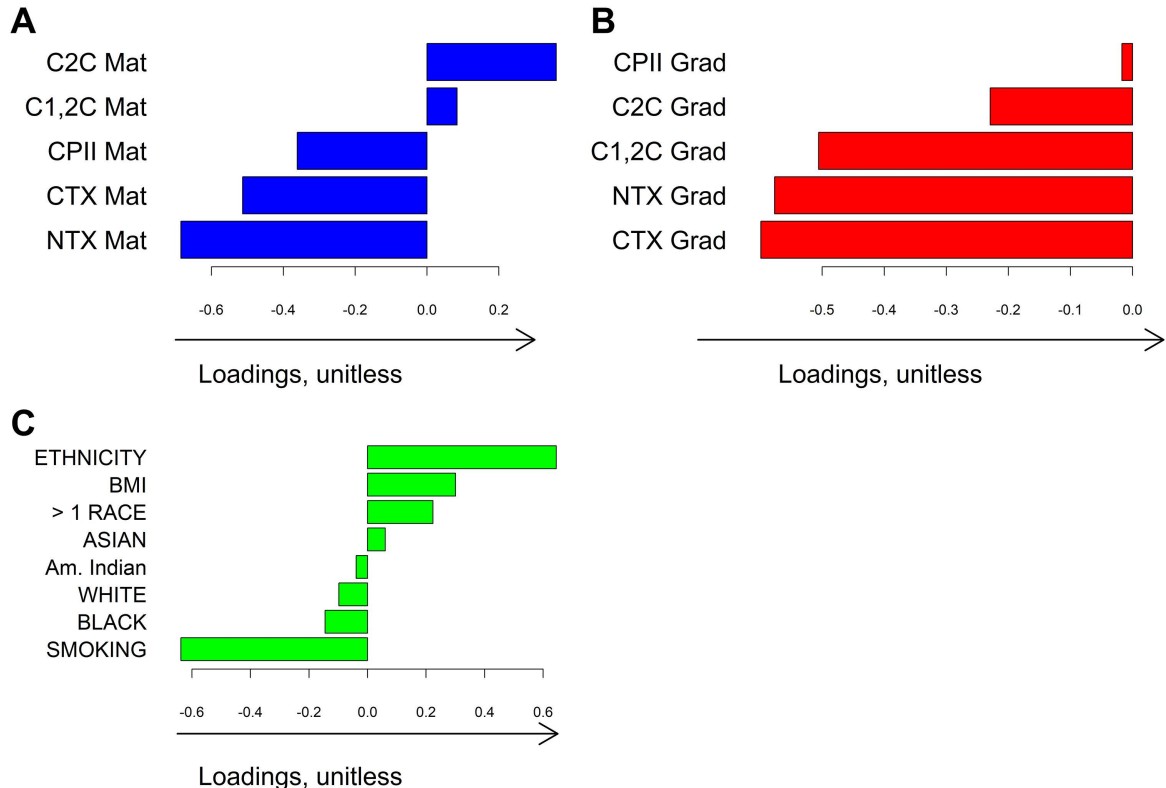

**Fig 2. First shared AJIVE direction of variation in males without knee ligament/meniscal injury at matriculation.** Feature loadings are represented by the bar plots indicating the amount that each biomarker and clinical variable contributed to the first shared direction. For all biomarkers at matriculation and graduation, age-adjusted values were computed as the residuals from linear multiple regression models of each biomarker on age. The loadings themselves are unitless and only have meaning qualitatively within the analysis such that a higher loading indicates greater contribution of the given variable in the corresponding mode of variation. **CTX:** C-terminal telopeptides of type I collagen; **NTX:** N-terminal telopeptides of type I collagen; **CPII:** C-terminal propeptide of Type II collagen; **C1,2C:** Collagen Type I and II collagenase-generated cleavage epitopes; **C2C:** Collagen Type II collagenase-generated cleavage epitopes**; BMI:** Body Mass Index; **Mat:** Matriculation; **Grad:** graduation.

variations in biomarkers at matriculation and graduation were depicted by the effect of age. Thus, for females, we processed the biomarker data without adjustment for age, while keeping age in block 3.

**Directions of individual variation: Biomarkers at matriculation.** We examined the components of individual variation for the block with biomarkers at matriculation. This block is of interest because of its potential to identify cadets at risk for injury. In males, we found three components representing individual variation in biomarker levels at matriculation (defined here as "Individual Components at matriculation", abbreviated as *ICm1, ICm2, and ICm3*, **Fig 4B**-**4D**). *ICm1* was characterized by high cartilage synthesis (CPII) and low cleavage of type I and II collagen; it indicates lower bone and cartilage degradation, suggesting a "protective" knee phenotype (**Fig 4B**). Males who experienced knee injuries during and after academy had lower individual *ICm1* scores (**Fig 4A**), suggesting that low cartilage synthesis prior to injury, in particular low CPII, can be an injury risk indicator. Confirming results from AJIVE, CPII levels measured at matriculation were higher in males without injuries during and after academy completion [mean=235.8, SD=121.6] compared to those with incident injuries [mean(SD): 195.04(118.1)] (p-value=0.05, t-test on age-adjusted levels of CPII) and were negatively associated with incident injuries after adjustment for smoking status (p=0.03, logistic regression).

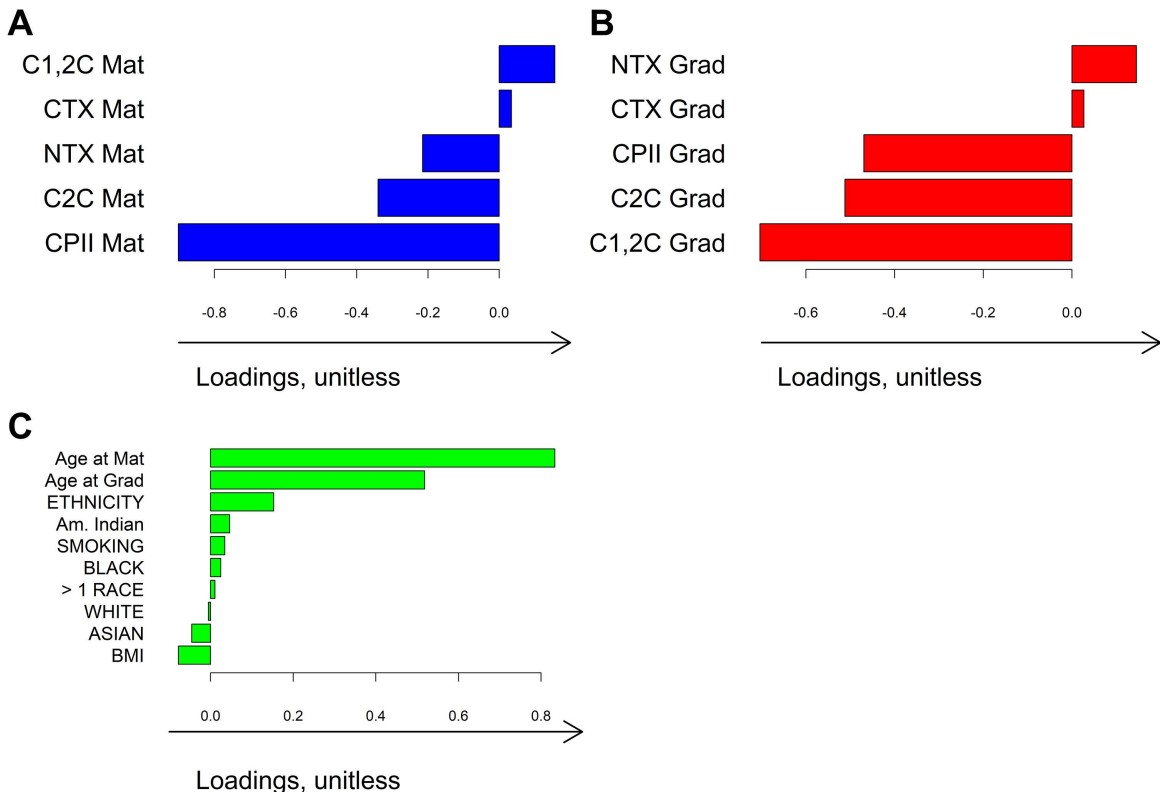

**Fig 3. First shared AJIVE direction of variation in females without knee ligament/meniscal injury at matriculation.** Feature loadings are represented by the bar plots indicating the amount that each biomarker and clinical variable contributed to the first shared direction. The loadings themselves are unitless and only have meaning qualitatively within the analysis such that a higher loading indicates greater contribution of the given variable in the corresponding mode of variation. **CTX:** C-terminal telopeptides of type I collagen; **NTX:** N-terminal telopeptides of type I collagen; **CPII:** C-terminal propeptide of Type II collagen; **C1,2C:** Collagen Type I and II collagenase-generated cleavage epitopes; **C2C:** Collagen Type II collagenase-generated cleavage epitopes**; BMI:** Body Mass Index; **Mat:** Matriculation; **Grad:** graduation.

In females, we also found three components of individual variation in the matriculation block. *ICm2* was similar to *ICm1* in males, suggesting a potential association of *ICm2* with incident injury. However, the distribution of scores for this and the two other components were similar in the two groups (with and without injuries after matriculation), **S3 Fig**.

### Secondary AJIVE analysis of data from participants with injury before, during, and after academy

Individuals injured before, during, and after the academy were included in the secondary analyses (81 males and 42 females), to explore individual modes of variation specific to biomarker levels at matriculation and graduation and for their association with post-academy KOOS scores.

**Shared variation.** The single component of shared variation in injured males (N = 81) was similar to that in our main sample (149 males without injuries at matriculation, **S4 Fig**). In injured women (N = 42), we found no modes of shared variation among the three data blocks.

**Individual variation.** In addition to the shared component, we examined the individual components of variation for each of the biomarker blocks (i.e., at graduation and at matriculation), which are independent of each other and of the demographic data. In males, there were three modes of variation in both biomarker blocks, with scores showing little difference between injured males who met the Luyten early OA criterion and those who did not. The individual structure

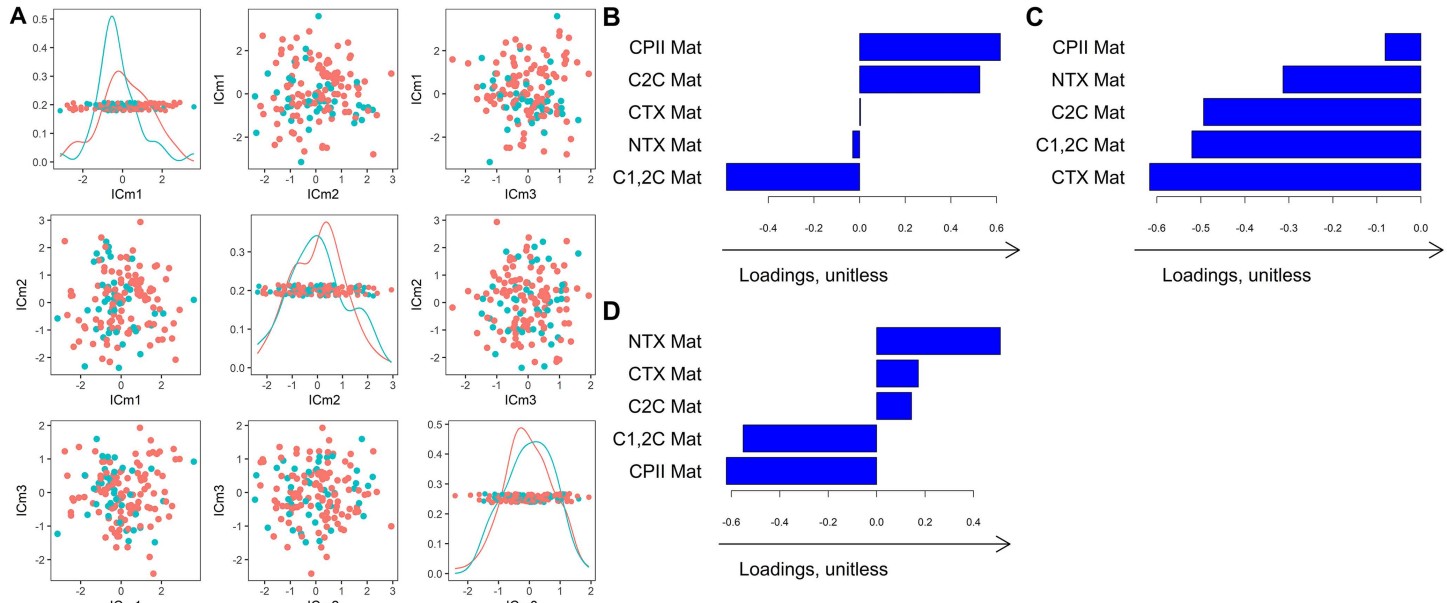

**Fig 4. First three directions representing individual variation in biomarker levels at matriculation (ICm1, ICm2, ICm3) in *males* that are *not* related to the biomarker levels at graduation and participant characteristics.** The red and blue curves on the diagonal plots (A) indicate the kernel density estimate for the distribution of the scores, colored by injury status after matriculation (injury = blue, no injury = red). The location of each individual in the coordinate system is defined by the individual directions shown in panels B, C, and **D.** Loading plots show the amount that each biomarker and clinical variable contributed to a given mode of variation. The loadings and scores are unitless and only have meaning qualitatively within the analysis such that a higher loading indicates greater contribution of the given variable in the corresponding mode of variation. **CTX:** C-terminal telopeptides of type I collagen; **NTX:** N-terminal telopeptides of type I collagen; **CPII:** C-terminal propeptide of Type II collagen; **C1,2C:** Collagen Type I and II collagenase-generated cleavage epitopes; **C2C:** Collagen Type II collagenase-generated cleavage epitopes; **Mat:** Matriculation.

of the graduation block ("Individual Components at graduation", *ICg)* suggested a potential "homeostatic cartilage" phenotype, represented by *ICg1* that was characterized by relatively low bone turnover and low cartilage breakdown and synthesis, **Fig 5B**. *ICg2* demonstrated high bone turnover and low cartilage breakdown (**Fig 5C**). Injured males who met the Luyten early OA illness KOOS criterion after academy had slightly lower ICg2 (i.e., lower bone turnover and higher cartilage breakdown), **Fig 5A**.

## Discussion

In these exploratory analyses, we confirmed known associations of joint metabolism biomarkers with age and smoking status among individuals without injury and provided new insights into the potential of these biomarkers to identify individuals at high risk of knee injury and knee problems consistent with early OA. We used data from the JUMP-ACL population which is relatively homogeneous with respect to physical activity and food intake at the time of the first serum acquisition (matriculation to the military academy), due to the physically demanding nature of the training and standard diet, which is an important advantage to our study. Demographic and clinical data, in combination with access to banked sera samples from pre-injury and post-injury time points, provide an exceptional opportunity to study the biological processes related to injury risk and post-injury outcomes over a relatively short span of time.

### Biomarker levels and demographic characteristics

Lower components of shared variation in males were associated with older age and lower concentrations of bone turnover markers, CTX and NTX, at matriculation and graduation, confirming previously known higher bone turnover in younger

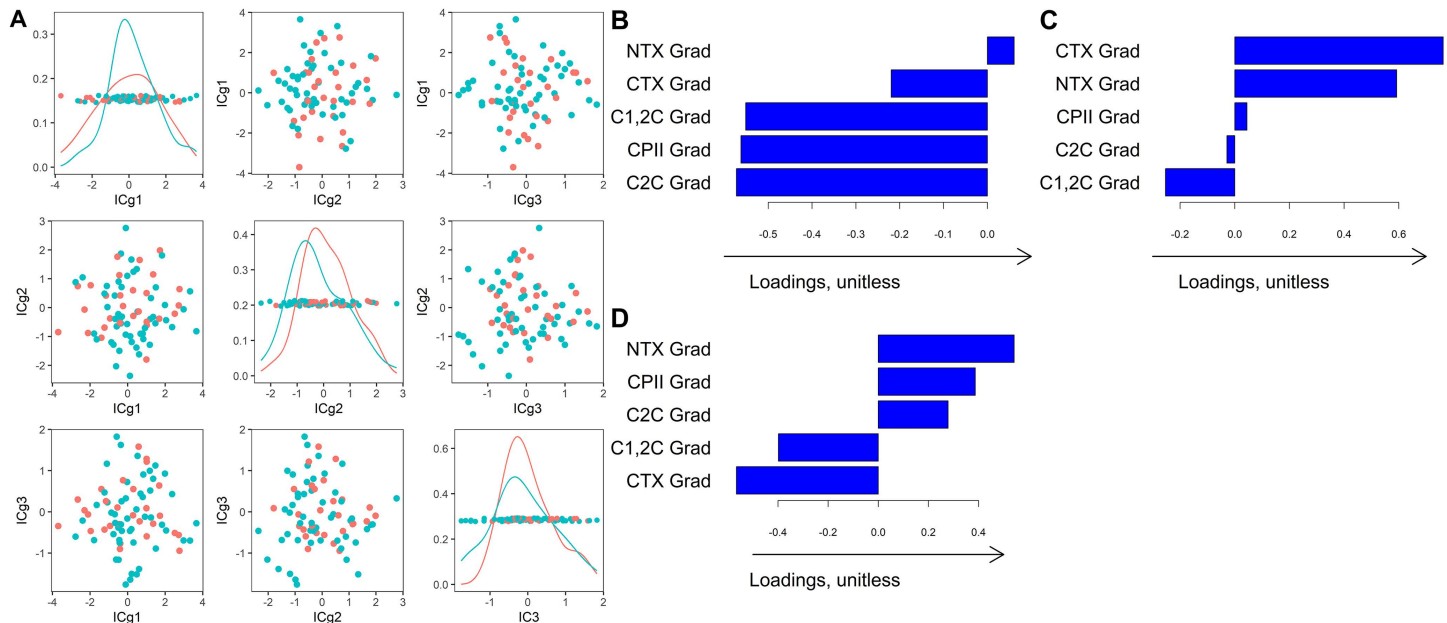

**Fig 5. First 3 directions representing individual variation in biomarker levels at graduation (ICg1, ICg2, ICg3) in *injured* males *not* related to the biomarker levels at matriculation and participant characteristics.** The red and blue curves on the diagonal plots (A) indicate the kernel density estimate for the distribution of the scores, colored by early OA illness status after graduation (meets Luyten criterion for a worse outcome = blue, does not meet criterion = red). The location of each individual in the coordinate system is defined by the individual directions shown in panels B, C, and **D.** Loading plots show the amount that each biomarker and clinical variable contributed to a given mode of variation. The loadings and scores are unitless and only have meaning qualitatively within the analysis such that a higher loading indicates greater contribution of the given variable in the corresponding mode of variation. **CTX:** C-terminal telopeptides of type I collagen; **NTX:** N-terminal telopeptides of type I collagen; **CPII:** C-terminal propeptide of Type II collagen; **C1,2C:** Collagen Type I and II collagenase-generated cleavage epitopes; **C2C:** Collagen Type II collagenase-generated cleavage epitopes; **Grad:** graduation.

adults and children due to lack of growth plate closure [33,34]. Interestingly, the components of shared variation in females showed no association between age and CTX and NTX levels (Fig 3). This can be explained by the fact that the peak of bone turnover biomarkers in females is reached at a younger age than in males (i.e., a few years before matriculation), plus peak values in young females are generally lower than in males [35,36]. In a previous study of young healthy females, CTX levels were higher in those with low BMI, whereas NTX concentrations were higher in current smokers [37]. In our study, the single component of shared variation in males was characterized by lower age-adjusted levels of CTX and NTX, higher BMI, and less smoking, confirming these associations. Loadings of the collagen synthesis marker CPII were negative at matriculation and positive at graduation in males, indicating a relative increase in CPII levels during academy training. Increased CPII levels were previously reported in female basketball players before playoffs, which may be a result of increased joint load during the regular season of active sports competition [38]. The loadings for the shared mode of variation in females indicate that serum C2C:CPII ratio levels were increased at graduation compared to matriculation suggesting an increase in type II collagen degradation relative to synthesis [39], which may also be related to joint overload, and/or decreased CPII related to growth plate closures.

## Biomarkers and risk of injury

An important advantage of the AJIVE method in the context of study with longitudinal biomarker assessments is the ability to interpret individual variations in biomarker levels at matriculation and graduation, which are not related to demographic data or to each other. The first component of individual variation in biomarkers levels at matriculation in males, or *ICm1,*

was characterized by high cartilage synthesis and low cleavage of type I and II collagen. In healthy cartilage, these processes are in balance, whereas an increased ratio of degradation over synthesis could cause cartilage loss, which may contribute to OA development [40]. In addition, CPII levels at matriculation were lower in males who experienced knee injury during and after the academy, which confirmed previous findings [21]. Therefore, *ICm1* is potentially a protective biomarker profile that needs to be further confirmed in independent studies.

### Biomarkers and post-injury symptoms

Due to the secondary analysis nature of our study design, which used existing measurements and serum samples from the DoDSR, KOOS data were not collected at specific timepoints from the injury. If KOOS was submitted after injury, the score may be influenced by the injury and not just by potential OA, however, all KOOS scores were obtained many years after injury (average of 7 years). Therefore, the findings from our secondary analyses involving injured men should be interpreted with caution and validated in future studies.

### Limitations

This was an exploratory study using a relatively small sample. These findings may not be generalizable to other young healthy populations without thorough replication in independent cohorts. In addition, although the military cohort is ideal for controlling for lifestyle factors, extrapolation of findings to broader OA populations is challenging. Differences in biomechanical loading patterns, access to healthcare, and injury mechanisms may limit applicability.

The time of day of the blood draw (morning vs. afternoon) and the detailed dietary information were not available in these data; therefore, we were unable to control for these determinants of biomarker levels. The timing of NTX and CTX collections is crucial due to the known circadian variations of these biomarkers, which may affect the results. This effect can be markedly reduced with an appropriate adjustment or when using fasting morning samples [41]. As some participants were in the age range corresponding to growth plate closure, we may have been limited in our ability to unequivocally establish that a relationship of a biomarker with OA was not due to a 'normal' age-related process independent of the OA process.

In females, although serum biomarkers vary with regard to type of contraceptive and duration of use, menstrual irregularity, and amenorrhea [42,43], this information was not available. This underscores the advisability, when feasible, of collecting information on hormone use by female study participants. In particular, it can be used as an additional block in the AJIVE approach, which would make it possible to distinguish biomarker changes associated with hormonal fluctuations from those associated with demographic and clinical data. *ICm1* in females was primarily characterized by lower CTX and NTX levels and was not associated with incident injury, similar to the other two individual biomarker modes of variation at matriculation, possibly due to small sample size.

### Conclusion

In conclusion, we confirmed known associations of joint metabolism biomarkers with age and smoking while providing new insights into the potential of these biomarkers, used singly (CPII) or in combination, to identify individuals at high risk of injury and post-injury outcomes. Furthermore, we proposed a framework that can be used in future studies of biomarkers and their trajectories in the context of injury-related knee OA in combination with demographic, clinical, genetic and omics data.

### Supporting information

**S1 Fig. Distribution of biomarker levels at matriculation and graduation after shifted log transformation.** The red and blue curves indicate the kernel density estimate for the distribution of biomarker levels colored by sex (male=red, female=blue). **CTX:** C-terminal telopeptides of type I collagen; **NTX:** N-terminal telopeptides of type I collagen;

CPII: C-terminal propeptide of Type II collagen; C12C: Collagen Type I and II collagenase-generated cleavage epitopes; C2C: Collagen Type II collagenase-generated cleavage epitopes; Mat: Matriculation; Grad: graduation.
(DOCX)

S2 Fig. First shared AJIVE direction of variation in MEN without knee ligament/meniscal injury at matriculation. X axes show the amount that each biomarker and clinical variable contributed to the first mode of variation. Biomarker levels at matriculation and graduation were transformed without adjustment on age. CTX: C-terminal telopeptides of type I collagen; NTX: N-terminal telopeptides of type I collagen; CPII: C-terminal propeptide of Type II collagen; C1,2C: Collagen Type I and II collagenase-generated cleavage epitopes; C2C: Collagen Type II collagenase-generated cleavage epitopes; BMI: Body Mass Index; Mat: Matriculation; Grad: graduation.
(DOCX)

S3 Fig. First 3 directions representing individual variation in biomarker levels at matriculation (ICm) in WOMEN that are not related to the biomarker levels at graduation and participant features.
(DOCX)

S4 Fig. First shared AJIVE direction of variation in injured MEN. CTX: C-terminal telopeptides of type I collagen; NTX: N-terminal telopeptides of type I collagen; CPII: C-terminal propeptide of Type II collagen; C1,2C: Collagen Type I and II collagenase-generated cleavage epitopes; C2C: Collagen Type II collagenase-generated cleavage epitopes; BMI: Body Mass Index; Mat: Matriculation; Grad: graduation.
(DOCX)

S1 File. Graphical abstract.
(PPTX)

## Acknowledgments

We would like to thank the participants in the JUMP ACL study, without whom this work would not be possible.

## Author contributions

**Conceptualization:** Virginia B. Kraus, Richard F. Loeser, Yvonne M. Golightly.

**Data curation:** Maryalice Nocera, Kenneth L. Cameron, Jesse R. Trump, Stephen W. Marshall, Yvonne M. Golightly.

**Formal analysis:** Liubov Arbeeva.

**Funding acquisition:** Amanda E. Nelson, Leigh F. Callahan, Stephen W. Marshall, Yvonne M. Golightly.

**Investigation:** Liubov Arbeeva, Virginia B. Kraus, Richard F. Loeser, Stephen W. Marshall, Yvonne M. Golightly.

**Methodology:** Liubov Arbeeva, Virginia B. Kraus, Richard F. Loeser, Yvonne M. Golightly.

**Project administration:** Maryalice Nocera, Kenneth L. Cameron, Jesse R. Trump, Stephen W. Marshall.

**Resources:** Kenneth L. Cameron, Jesse R. Trump, Stephen W. Marshall.

**Software:** Liubov Arbeeva.

**Supervision:** Virginia B. Kraus, Yvonne M. Golightly.

**Validation:** Liubov Arbeeva.

**Visualization:** Liubov Arbeeva.

**Writing – original draft:** Liubov Arbeeva, Virginia B. Kraus, Yvonne M. Golightly.

**Writing – review & editing:** Amanda E. Nelson, Maryalice Nocera, Leigh F. Callahan, Richard F. Loeser, Kenneth L. Cameron, Jesse R. Trump, Stephen W. Marshall.

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
