## [Decision Letter · Decision Letter 0]

17 Nov 2025

Dear Dr. Arbeeva,

Thank you for submitting your manuscript to PLOS ONE. After careful consideration, we feel that it has merit but does not fully meet PLOS ONE’s publication criteria as it currently stands. Therefore, we invite you to submit a revised version of the manuscript that addresses the points raised during the review process.

We look forward to receiving your revised manuscript.

Kind regards,

Santhi Silambanan, MD, DNB

Academic Editor

PLOS ONE

Journal Requirements:

Financial Support: National Institute of Arthritis and Musculoskeletal and Skin Diseases P60AR064166, P30AR072580

5. Thank you for stating the following in the Competing Interests/Financial Disclosure section:

AEN reports honoraria from Novo Nordisk, support for travel from Osteoarthritis Research Society International and American College of Rheumatology as well as grant funding from NIH/NIAMS not related to this work.

Other authors have declared that no competing interests exist.

We note that one or more of the authors are employed by a commercial company: Novo Nordisk

6. In this instance it seems there may be acceptable restrictions in place that prevent the public sharing of your minimal data. However, in line with our goal of ensuring long-term data availability to all interested researchers, PLOS’ Data Policy states that authors cannot be the sole named individuals responsible for ensuring data access (http://journals.plos.org/plosone/s/data-availability#loc-acceptable-data-sharing-methods).

Additional Editor Comments:

Dear authors please respond to the queries raised by the reviewers

Reviewers' comments:

Reviewer's Responses to Questions

**Comments to the Author**

1. Is the manuscript technically sound, and do the data support the conclusions?

Reviewer #1: Yes

Reviewer #2: Partly

Reviewer #3: Yes

2. Has the statistical analysis been performed appropriately and rigorously?

Reviewer #1: I Don't Know

Reviewer #2: I Don't Know

Reviewer #3: No

3. Have the authors made all data underlying the findings in their manuscript fully available?

Reviewer #1: Yes

Reviewer #2: No

Reviewer #3: Yes

4. Is the manuscript presented in an intelligible fashion and written in standard English?

Reviewer #1: Yes

Reviewer #2: Yes

Reviewer #3: No

Reviewer #1: I accept this article without changes. The methodology and the discussions seem to be logical with the biomarker discussed. I am not sure about the statistical analysis part of the manuscript and therefore has to be reviewed for the same.

Reviewer #2: 1. Kindly provide details of the study design of the current paper.

2. Introduction section is lengthy. It needs to be made precise.

3. Kindly follow instructions to authors. The paper does not comply with it. Kindly do the needful.

4. The methods section does not provide sufficient scientific information.

Reviewer #3: This is an interesting study that presents a longitudinal analysis of joint metabolism serum biomarkers

Introduction

This is appropriately discussed. The is proper balance between the general info and the more specific info outlining the specific purpose.

Line 57-85. Please consider focusing more on the post-traumatic OA

Line 86-92. The readers could benefit from a short description of the current knowledge on the biomarkers and their association with OA

Methods

This part is the strongest of the manuscript, with clear descriptions and data collection

Line 109-131. Very clear description of the study and its phases.

Line 135-158. Please present clearly when these specimens obtained.

Line 165-172. This part needs to be more clear. When were these collected? What happened to the patients that had an injury and what was the treatment? Were KOOS scores measured at specific timepoints from the injury/surgery if needed.

Line 218-224. Additional support or further explanation with some literature support may be useful for the analysis of high and low risk. Some of these data would be helpful to be presented as raw data.

Line 260-272. This part needs to be presented in a clear manner and some data on the patient with injury and without injury separately.

Line 279-296. Specific statistical analysis values are necessary. It is hard to evaluate slightly lower without a specific value.

Line 266-268. Specific data association with KOOS scores are important

Discussion

More clear and specific discussion may be possible

Appropriately discussed limitations

**Do you want your identity to be public for this peer review?** For information about this choice, including consent withdrawal, please see our Privacy Policy

Reviewer #1: **Yes:**  Dr.Jomon De Joseph., MBBS, DNB, MRCS(Glasgow)

Reviewer #2: No

Reviewer #3: No

---

## [Author Response · Author response to Decision Letter 1]

19 Dec 2025

EDITOR

1. Formatting and Style Compliance

We have ensured that the manuscript now fully adheres to PLOS ONE’s style requirements, including file naming conventions, as per the templates provided.

2. ORCID iD Validation

The corresponding author’s ORCID iD has been validated in Editorial Manager.

3. Funding Information

We double-checked the grant numbers in the ‘Funding Information’ section to ensure they accurately reflect the awards received for this study. Only 2 grants (P60AR064166, P30AR072580) supported the work on this manuscript. Other grant numbers were outlined in Competing Risk Statements as they were not related to this work. Therefore, we did not change this part.

4. Role of Funders

The amended statement regarding the role of funders is as follows:

5. Funding and Competing Interests Statements

We have updated both statements to reflect the commercial affiliation and clarify roles:

Funding Statement:

“Financial support was provided by the National Institute of Arthritis and Musculoskeletal and Skin Diseases (P60AR064166, P30AR072580). The funder provided support in the form of salaries for authors but did not have any additional role in the study design, data collection and analysis, decision to publish, or preparation of the manuscript. The specific roles of these authors are articulated in the ‘Author Contributions’ section.”

Competing Interests Statement:

“AEN reports honoraria from Novo Nordisk, MedScape Education, and CCR-West 2024, support for travel from NYU Langone, Hospital for Special Surgery, Osteoarthritis Research Society International, and American College of Rheumatology as well as grant funding from NIH/NIAMS not related to this work (R01AR078187; R01AR080742; K24AR081368; R01AR077060; R01AR080733). YMG reports grant funding from NIH/NIAMS not related to this work (R34AR083077; R01AR078187; R01AR080742; R01AR080733; R56AR080060; R21AR080309). SWM reports grant funding from NIH/NIAMS not related to this work (R01 AR050461). This does not alter our adherence to PLOS ONE policies on sharing data and materials.”

6. Data Availability

Below is non-author institutional contact information for data access requests:

Institutional Review Boards of the University of North Carolina at Chapel Hill (UNC-CH 13-0059) and the Keller Army Community Hospital (KACH 15-007).

We did not provide Name of Contact due to common staff changes, but all questions can be sent to irb_questions@unc.edu email

Address: 104 Airport Drive, Suite 2100, CB#7097, Chapel Hill, NC 27599

Telephone: (919) 966 3113

Additionally, we have implemented measures to ensure persistent and long-term data storage and availability. All study data are securely stored on UNC-protected servers, and in the event that all authors become unavailable, UNC IT personnel will maintain access and provide data upon request.

7. References

The reviewers did not recommend any specific references. However, to address their concerns and clarify the points raised, we have added relevant citations that we identified as appropriate and supportive of our revisions

Reviewer #1:

I accept this article without changes. The methodology and the discussions seem to be logical with the biomarker discussed. I am not sure about the statistical analysis part of the manuscript and therefore has to be reviewed for the same.

Author response: Thank you!

Reviewer #2:

1. Kindly provide details of the study design of the current paper.

Author response: Thank you for your comment. We have revised the manuscript to make the study design more explicit. Specifically, we have added a separate subsection titled Study Design at the beginning of the Methods section. This subsection clearly states that the current analysis is a secondary analysis of two linked studies (JUMP-ACL and JUMP-ACL OA) and summarizes the overall design and purpose.

We have retained the detailed description of participant enrollment and flow in the Participants subsection and referenced Figure 1, which visually presents the study design and participant selection process. We hope these changes address your concern and improve clarity for readers.

Changes made: added a separate subsection titled Study Design at the beginning of the Methods section. Additional edits to clarify.

2. Introduction section is lengthy. It needs to be made precise.

Author response/ Changes made:

Thank you for your comment. To address it, along with Reviewer 3’s suggestion to focus more on post-traumatic OA, we have revised the introduction to make it more precise and aligned with the study objectives.

3. Kindly follow instructions to authors. The paper does not comply with it. Kindly do the needful.

Author response/ Changes made: Thank you for your comment. We received similar feedback from the editor, which included more specific instructions regarding compliance with PLOS ONE’s author guidelines. We have carefully revised the manuscript to address both the editor’s detailed requests and your concern simultaneously.

We hope these changes have improved the manuscript and addressed your concern regarding compliance with the journal’s instructions to authors.

These revisions include:

1) Ensuring the manuscript follows PLOS ONE’s style requirements and formatting templates.

2) Validating the corresponding author’s ORCID iD in Editorial Manager.

3) Correcting and aligning grant information in the Funding Information and Financial Disclosure sections.

4) Updating the Role of Funder statement and including it in the cover letter.

5) Amending the Funding Statement and Competing Interests Statement to accurately reflect commercial affiliations and confirm adherence to PLOS ONE policies.

6) Providing non-author institutional contact information for data access in compliance with PLOS ONE’s Data Policy.

7) Reviewing and evaluating any suggested citations for relevance.

4. The methods section does not provide sufficient scientific information.

Author response/ Changes made: We appreciate your feedback regarding the Methods section. While we acknowledge your concern that the section may lack sufficient scientific information, we also received more detailed and constructive comments from Reviewer 3, who considered this section the strongest part of the manuscript and provided specific suggestions for further improvement.

We incorporated Reviewer 3’s recommendations, such as clarifying specimen collection timing, patient treatment details, and KOOS score measurement timepoints, to further strengthen the section. We also clarified that in our main analysis we included only the individuals without prevalent injury reported at matriculation. In the exploratory analysis, we used only data from individuals with injuries (including those who were in main analysis AND those who were excluded due to prevalent injuries).

Reviewer #3:

This is an interesting study that presents a longitudinal analysis of joint metabolism serum biomarkers

Author response: We appreciate your comment.

Introduction

This is appropriately discussed. The is proper balance between the general info and the more specific info outlining the specific purpose.

Author response: Thank you!

Line 57-85. Please consider focusing more on the post-traumatic OA

Author response/ Changes made: Thank you, we revised the Introduction as you suggested.

Line 86-92. The readers could benefit from a short description of the current knowledge on the biomarkers and their association with OA

Author response/ Changes made: We included a brief overview of current OA biomarker research; additional details are available in a recent comprehensive review, reference 12.

Methods

This part is the strongest of the manuscript, with clear descriptions and data collection

Author response: Thank you!

Line 109-131. Very clear description of the study and its phases.

Author response: Thank you!

Line 135-158. Please present clearly when these specimens obtained.

Author response/ Changes made: Thank you for this comment, additional details are provided (lines 141-145).

Line 165-172. This part needs to be more clear. When were these collected? What happened to the patients that had an injury and what was the treatment? Were KOOS scores measured at specific timepoints from the injury/surgery if needed.

Author response: Thank you. We have revised this section.

Changes made: We described these details as follows: “ACL and knee injuries of comparable severity that were experienced as a military cadet, or as an active duty officer, underwent surgical repair and post-surgical rehabilitation from experienced orthopedic and post-surgical team, with the goal of restoring full pre-injury function to the fullest extent possible. If surgery is not an option, then injuries were managed and rehabilitated using best possible surgical techniques.

The KOOS items were administered between 2015 and 2017, therefore they were obtained at a variable number of years after injury, with an average of 7 years post-injury”.

Line 218-224. Additional support or further explanation with some literature support may be useful for the analysis of high and low risk. Some of these data would be helpful to be presented as raw data.

Author response/ Changes made:

We appreciate the reviewer’s comment and agree that clarification was needed. The confusion resulted from our initial description of the study population. The main analysis was restricted to cadets without injury at matriculation, who could be at either higher or lower risk for injury during training. The biomarkers evaluated may help identify those at higher risk. To address this, we have revised the last two paragraphs of the Methods section to clarify the inclusion criteria and the rationale for analyzing high- and low-risk groups, see also the next comment and our response. We apologize if we misunderstood the comment.

Line 260-272. This part needs to be presented in a clear manner and some data on the patient with injury and without injury separately.

Author response/ Changes made:

We appreciate your suggestion. We have revised this section by adding subheadings to distinguish results for participants with injury from those without injury at matriculation. We also included the number of participants (N) in each group to improve clarity. We also clarified at the end of the methods section as follows: “All main analyses identifying individuals at risk included only participants without prevalent injuries at matriculation. Additional exploratory analyses were performed on individuals excluded from the main analysis due to prevalent injuries at matriculation combined with those from the main analysis who sustained an injury during and after the academy (e.g. individuals injured before, during, and after the academy) to explore whether biomarker levels identify individuals at high risk of post-injury symptoms”.

Line 279-296. Specific statistical analysis values are necessary. It is hard to evaluate slightly lower without a specific value.

Author response: Thank you and we agree. We added mean (SD) in males with and without injuries.

Changes made: The sentence “Confirming results from AJIVE, age-adjusted levels of CPII measured at matriculation were higher in males without injuries during and after academy completion (p-value=0.05, t-test) and were negatively associated with incident injuries after adjustment for smoking status (p=0.03, logistic regression)” was revised as follows: “Confirming results from AJIVE, CPII levels measured at matriculation were higher in males without injuries during and after academy completion [mean=235.8, SD=121.6] compared to those with incident injuries [mean(SD): 195.04(118.1)] (p-value=0.05, t-test on age-adjusted levels of CPII) and were negatively associated with incident injuries after adjustment for smoking status (p=0.03, logistic regression)”.

Line 266-268. Specific data association with KOOS scores are important

Author response: The Knee Injury and Osteoarthritis Outcome Score (KOOS) is a self administered questionnaire developed to assess short and long term patient relevant outcomes following knee injury. In our main analysis, we included cadets without prevalent injury at matriculation, some of whom may have sustained injuries later. Lines 266-268 describe the results from our main analysis. We did not compare KOOS scores in the main analyses because (1) they are less relevant for individuals who remained uninjured after matriculation, and (2) KOOS was not collected at a standardized time point following injury. We acknowledged this limitation in discussion.

Discussion

More clear and specific discussion may be possible

Appropriately discussed limitations

Author response/ Changes made: Thank you, we revised this section.

Dear Editor,

We thank PLOS ONE for considering our manuscript entitled, " A Novel Approach for Longitudinal Analysis of Serum Biomarkers of Joint Metabolism and Knee Injury in Military Officers" (Number PONE-D-25-49194). We also would like to thank the reviewers and deeply appreciate their insightful comments and suggestions.

Please find bellow our point-by-point responses and the revised manuscript accordingly.

We hope that you will consider it for publication in your journal.

Sincerely,

Liubov Arbeeva

EDITOR

1. Formatting and Style Compliance

We have ensured that the manuscript now fully adheres to PLOS ONE’s style requirements, including file naming conventions, as per the templates provided.

2. ORCID iD Validation

The corresponding author’s ORCID iD has been validated in Editorial Manager.

3. Funding Information

We double-checked the grant numbers in the ‘Funding Information’ section to ensure they accurately reflect the awards received for this study. Only 2 grants (P60AR064166, P30AR072580) supported the work on this manuscript. Other grant numbers were outlined in Competing Risk Statements as they were not related to this work. Therefore, we did not change this part.

4. Role of Funders

The amended statement regarding the role of funders is as follows:

5. Funding and Competing Interests Statements

We have updated both statements to reflect the commercial affiliation and clarify roles:

Funding Statement:

“Financial support was provided by the National Institute of Arthritis and Musculoskeletal and Skin Diseases (P60AR064166, P30AR072580). The funder provided support in the form of salaries for authors but did not have any additional role in the study design, data collection and analysis, decision to publish, or preparation of the manuscript. The specific roles of these authors are articulated in the ‘Author Contributions’ section.”

Competing Interests Statement:

“AEN reports honoraria from Novo Nordisk, MedScape Education, and CCR-West 2024, support for travel from NYU Langone, Hospital for Special Surgery, Osteoarthritis Research Society International, and American College of Rheumatology as well as grant funding from NIH/NIAMS not related to this work (R01AR078187; R01AR080742; K24AR081368; R01AR077060; R01AR080733). YMG reports grant funding from NIH/NIAMS not related to this work (R34AR083077; R01AR078187; R01AR080742; R01AR080733; R56AR080060; R21AR080309). SWM reports grant funding from NIH/NIAMS not related to this work (R01 AR050461). This does not alter our adherence to PLOS ONE policies on sharing data and materials.”

6. Data Availability

Below is non-author institutional contact information for data access requests:

Institutional Review Boards of the University of North Carolina at Chapel Hill (UNC-CH 13-0059) and the Keller Army Community Hospital (KACH 15-007).

We did not provide Name of Contact due to common staff changes, but all questions can be sent to irb_questions@unc.edu email

Address: 104 Airport Drive, Suite 2100, CB#7097, Chapel Hill, NC 27599

Telephone: (919) 966 3113

Additionally, we have implemented measures to ensure persistent and long-term data storage and availability. All study data are securely stored on UNC-protected servers,

---

## [Editor Report · Decision Letter 1]

13 Jan 2026

A Novel Approach for Longitudinal Analysis of Serum Biomarkers of Joint Metabolism and Knee Injury in Military Officers

PONE-D-25-49194R1

Dear Dr. Arbeeva,

We’re pleased to inform you that your manuscript has been judged scientifically suitable for publication and will be formally accepted for publication once it meets all outstanding technical requirements.

Kind regards,

Santhi Silambanan, MD, DNB

Academic Editor

PLOS One

Additional Editor Comments (optional):

The authors have responded to the queries raised by the three reviewers. The responses were appropriately incorporated in the revised manuscript.
---

## [Editor Report · Acceptance letter]

PONE-D-25-49194R1

PLOS One

Dear Dr. Arbeeva,

I'm pleased to inform you that your manuscript has been deemed suitable for publication in PLOS One. Congratulations! Your manuscript is now being handed over to our production team.

Kind regards,

on behalf of

Dr. Santhi Silambanan

Academic Editor

PLOS One